# Protection Against Pneumonia Induced by Vaccination with Fimbriae Subunits from *Klebsiella pneumoniae*

**DOI:** 10.3390/vaccines13030303

**Published:** 2025-03-11

**Authors:** Lucas Assoni, Isabelle Ciaparin, Monalisa Martins Trentini, Juliana Baboghlian, Gabriel Rodrigo, Brenda Vieira Ferreira, José Aires Pereira, Carlos Martinez, Lucio Ferraz, Raquel Girardello, Lucas Miguel Carvalho, Anders P. Hakansson, Thiago Rojas Converso, Michelle Darrieux

**Affiliations:** 1Laboratório de Microbiologia Molecular e Clínica, Universidade São Francisco, Bragança Paulista 12916-900, Brazilisabelle.ciaparin@mail.usf.edu.br (I.C.); juliana.baboghlian@mail.usf.edu.br (J.B.); gabriel.rodrigo@mail.usf.edu.br (G.R.); brenda.vieira@mail.usf.edu.br (B.V.F.); lucio.ferraz@usf.edu.br (L.F.); raquel.girardello@usf.edu.br (R.G.); 2Laboratório Especial de Desenvolvimento de Vacinas, Instituto Butantan, São Paulo 05508-040, Brazil; monalisa.trentini.esib@esib.butantan.gov.br; 3Laboratório de Investigações Médicas, Universidade São Francisco, Bragança Paulista 12916-900, Brazil; jose.pereira@usf.edu.br (J.A.P.); carlos.martinez@usf.edu.br (C.M.); 4Laboratório de Biologia de Sistemas e Ômicas em Ciências da Saúde (LaBSOmiCS), Universidade São Francisco, Bragança Paulista 12916-900, Brazil; lucas.miguel@usf.edu.br; 5Division of Experimental Infection Medicine, Department of Translational Medicine, Lund University, 222 42 Lund, Sweden; anders_p.hakansson@med.lu.se

**Keywords:** *K. pneumoniae*, fimbriae, protein vaccines, biofilms

## Abstract

Background: *Klebsiella pneumoniae* infections pose a great burden worldwide, causing high morbidity and mortality, which are worsened by the increase in multidrug-resistant strains. New therapeutic/prophylactic strategies are urgently needed to overcome antibiotic resistance and reduce the health and economic impacts of diseases caused by this pathogen. Fimbriae are important virulence factors involved in biofilm formation and adhesion to host cells. Their exposed location, conservation among clinical isolates and adjuvant properties make them interesting candidates for inclusion in protein-based vaccines. Therefore, the present work investigated the immunological potential of type 1 and 3 fimbriae subunits in a murine model of *K. pneumoniae* lung infection. Methods: MrkA and FimA were produced as recombinant proteins in *E. coli*, purified and used to immunize mice subcutaneously. The immune responses were characterized and protection against pneumonia was evaluated after intranasal challenge. Results: Subcutaneous immunization with recombinant FimA and MrkA induced high IgG1 production; the antibodies efficiently recognized the native proteins at the bacterial surface, promoted C3 deposition and reduced biofilm formation by *K. pneumoniae* in vitro. Mice vaccinated with the co-administered proteins reduced the bacterial loads in the lungs after intranasal challenge, less inflammation and reduced tissue damage. Conclusion: The results suggest that both type 1 and type 3 fimbriae contribute to protection against *K. pneumoniae* lung infection, inducing antibodies that bind to the bacteria and favoring Complement deposition and clearance by the host, while inhibiting biofilm formation.

## 1. Introduction

Despite the efforts of health agencies worldwide to reduce the spread of infections caused by antimicrobial-resistant bacteria, such infections still represent a major threat. Among the multidrug-resistant pathogens, *K. pneumoniae* is currently regarded as the number 1 priority by the WHO, reinforcing the need for new therapeutic and or prophylactic measures [1].

*K. pneumoniae* is a Gram-negative bacterium that commonly colonizes mucous membranes, such as those of the respiratory, gastrointestinal and urinary tracts [2,3]. It is associated with pneumonia, urinary tract infections, sepsis, hepatic and non-hepatic abscesses, meningitis, endophthalmitis [3] and fasciitis, the latter being mainly related to hypervirulent strains [4]. *Klebsiella* species are among the leading causes of healthcare-associated infections, with a mortality rate of 42.14%, which is significantly higher in patients admitted to the ICU [5]; it is also the third most frequently isolated species in cases of pneumonia [6].

In intensive care units, ventilator-associated pneumonia (VAP) remains a major concern, with multidrug-resistant *K. pneumoniae* among the leaders in terms of mortality [7]. Previous studies suggest that VAP caused by *K. pneumoniae* accounts for 11.9–37% of total cases [8,9]. Although less common than other pathogens, there is a growing rate of colonization and respiratory infections by *K. pneumoniae* in the community, mainly due to hypervirulent strains (HMV+) [10,11]. These strains exhibit enhanced capsule production, which significantly increases their resistance to host clearance mechanisms [12,13], further complicating infection control.

Despite the significant burden associated with *K. pneumoniae*, there are no vaccines available against systemic and respiratory infections by this pathogen. Throughout the years, different vaccine formulations have been proposed and few reached clinical trials [14]. The main types of formulations tested against this pathogen were bacterial lysates and capsular components, which have limitations related to their safety and/or vaccine coverage [15]. One promising alternative is the use of protein-based vaccines, which are more conserved antigens able to elicit effective immune responses with broad coverage.

Fimbriae are key elements in the pathogenesis of *K. pneumoniae*. They promote bacterial attachment to abiotic surfaces and host tissues, such as the mucosa of lung or urinary tract epithelium, leading to colonization, invasion and biofilm formation [16,17]. *K. pneumoniae* biofilms represent an important source of infection, being associated with the contamination of medical devices such as intraurethral catheters and ventilators, which favor bacterial spread and impair antibiotic treatment [17].

The two main types of fimbriae expressed by *K. pneumoniae* are type 1, coded by the *fim* gene cluster (*fimAICDFGHK*), under the regulation of *fimS*, and type 3, including the *mrk* operon (*mrkABCDF*), which is under the regulation of the *mrkHIJ* cluster. Type 1 fimbriae are filamentous protein structures including FimH and FimA protein subunits. FimH acts by facilitating lectinophagocytosis, a type of phagocytosis carried out by macrophages and neutrophils. In addition, it attaches to immune cells such as mast cells, stimulating immune activation. This process increases the recruitment of neutrophils to the site of infection, which contributes to bacterial clearance [17]. FimA is the main component of type 1 fimbriae, forming a hair-like, polymeric structure responsible for the adhesion to and invasion of bladder cells, as well as contributing to biofilm formation on these cells and on abiotic surfaces [18].

Type 3 fimbriae are helix-shaped structures composed of an adhesin, MrkD, which interacts with the extracellular matrix, and polymers of MrkA, forming the fimbriae shaft. MrkA can also bind to abiotic surfaces, leading to biofilm formation [17]. In addition, type 3 fimbriae increase the production of ROS by neutrophils, contributing to oxidative damage within the host.

Their central role in bacterial attachment and biofilm formation, combined with exposure at the bacterial surface and conservation among strains, make fimbriae components interesting vaccine candidates. Therefore, the present study evaluated the effects of immunization with recombinant FimA and MrkA in a mouse lung colonization model. Vaccine-induced antibodies were tested for their ability to recognize the native proteins on the bacterial surface, promote Complement deposition and inhibit biofilm formation on abiotic surfaces. The inflammatory response and tissue damage were also evaluated in immunized versus control mice and correlated with the vaccine response and protection against pneumonia.

## 2. Materials and Methods

### 2.1. Bacterial Strains and Growth Conditions

The genes encoding FimA and MrkA were cloned from *K. pneumoniae* ATCC 10031 strain. The lung colonization challenge and the functional studies were performed with the BM567 strain, a K2-serotype ESBL-producing clinical isolate obtained from a bloodstream infection at Hospital Universitário São Francisco de Assis. This strain was selected using in vitro biofilm formation and fimbriae expression assays (Appendix A) Fimbriae expression in BM567 was confirmed using the in vitro yeast agglutination assay, as described by Pacheco et al. (2021) [19].

### 2.2. Protein Production

Recombinant FimA and MrkA were produced as recombinant proteins in *E. coli* Briefly, the *fimA* and *mrkA* genes were amplified from ATCC 10031 strain and subsequently cloned in the pET-28a (Novagen—Darmstadt, Germany) and pAE-6xHis [20] vectors, respectively. The recombinant vectors, pET_*fimA* and pAE_*mrkA*, were then transformed into the BL21 Star and BL21 (Salt-Induced) *Escherichia coli* strains for recombinant protein expression. The recombinant proteins were purified by affinity chromatography using HisTrap 5 mL columns (Cytiva—Marlborough, MA, USA). After elution, proteins underwent dialysis, quantification, and LLPS removal using Triton X-114, as described by [21]. The protein concentration was determined using the Bradford reagent assay (LGC Biotecnologia—Cotia, Brazil)

### 2.3. Ethics Statements

The animal experimentation described in this study was approved by the ethics committee at São Francisco University, Bragança Paulista, Brazil (protocol number 008.11.2020), and performed in accordance with the relevant guidelines. C57BL/6 mice were randomly distributed in groups of five animals per cage, with filtered air and controlled temperature and luminosity. Food and water were supplied ad libitum. At the end of the experiments, all surviving mice were humanely euthanized using a mixture of ketamine and xylazine at three times the concentration required for anesthesia.

### 2.4. Mouse Immunization and Challenge

The animals (7 weeks old) were immunized subcutaneously with three doses of 10 µg of rFimA or rMrkA (or 10 µg of each protein in the co-administered group) in ten-day intervals, with 100 µg of aluminum hydroxide (Al(OH)_3_) as the adjuvant, diluted in 0.9% saline (final immunization volume of 100 µL). The control group (sham immunized mice) received 100 µg of adjuvant diluted in saline. Seven days after the last dose, blood was collected via retro-orbital puncturing and the sera, separated by centrifugation and stored at −20 °C.

Ten days after the last immunization, the mice were challenged intranasally with 1.8 × 10^7^ CFU suspension of *K. pneumoniae* BM567, diluted in 50 µL per mouse. Prior to inoculation, the animals were anesthetized with a ketamine–xylazine mixture to allow for aspiration of the inoculum directly into the lungs. The mice were monitored for morbidity signs and 48 h following inoculation, they were euthanized and the lungs were aseptically collected and homogenized with a syringe plunger against a 100 µm cell strainer (Corning—Corning, NY, USA) in 1 mL of cold PBS. Serial dilutions of the lung homogenates were plated in LB agar supplemented with 2.5 µg/mL of gentamicin to count CFUs.

One animal from each group was randomly selected for histopathological analysis of the lungs. A comparative histological analysis was conducted among healthy (unchallenged) controls, challenged controls and immunized/challenged mice; inflammatory and tissue damage scores of the histological slides were graded as absent (-), slight (+), moderate (++) or intense (+++). To prevent biases, this analysis was performed by two independent researchers.

### 2.5. Antibody Production and Isotyping

Production of IgG1 and IgG2a was measured by ELISA as follows: The plates were coated with 2 µg/mL of the recombinant proteins in a high-binding, flat-bottom, 96-well plate (Corning) and sera of the immunized animals were diluted serially. An IgG calibration curve was generated using unlabeled mouse IgG1 or IgG2A (Southern Biotech Biotech—Birmingham, AL, USA). The detection of IgG1 and IgG2A subclasses was carried out using a goat anti-mouse IgG1 or anti-IgG2A antibodies (Southern Biotech), followed by a secondary incubation with rabbit anti-goat IgG conjugated to HRP (Southern Biotech). Following the washing steps, o-phenylenediamine dihydrochloride (OPD) (Sigma-Aldrich—St. Louis, MO, USA) was added to the plate for color development. The absorbance was measured at A_492nm_ using an Expert Plus plate reader (Asys—Holliston, MA, USA). The antibody concentration in the sera was calculated by interpolating the absorbance values against the standard curve, followed by a linear regression analysis of the sample values.

### 2.6. IgG Binding Assay

*K. pneumoniae* BM567 was grown in Luria–Bertani (LB) broth to a concentration of approximately 3.5 × 10^8^ CFU/mL, centrifuged, washed once with PBS and subsequently suspended in Hank’s balanced salt solution buffer (HBSS), supplemented with 0.1% bovine gelatin. The bacteria were incubated with heat-inactivated sera from immunized mice (with the final antibody concentration adjusted to 0.06 mg/mL through the dilution of anti-MrkA in control serum) in 100 µL for 30 min, at 37 °C in a “U” bottom 96-well plate (Corning). After incubation, the bacteria were washed with PBS and then incubated with goat anti-mouse IgG conjugated with FITC (fluorescein isothiocyanate, MP Biomedicals—Santa Ana, CA, USA) at a final concentration of 1:500. The samples were washed twice, resuspended with 200 µL of 2% paraformaldehyde and the fluorescence, measured using a FACS Canto II flow cytometer (BD Biosciences Franklin Lakes, NJ, USA). Data analysis was performed using FlowJo software (version 10.10.0).

### 2.7. C3 Deposition on K. pneumoniae

*K. pneumoniae* grown in Luria–Bertani (LB) broth to 3.5 × 10^8^ CFU/mL washed once with PBS and subsequently suspended with Hank’s balanced salt solution buffer (HBSS), supplemented with 0.1% bovine gelatin. Next, the bacteria were incubated with inactivated sera from vaccinated and control mice at a final concentration of 0.06 mg/mL in 100 µL for 30 min, at 37 °C in a “U” bottom 96-well plate (Corning). Normal mouse serum (NMS) was added as a complement source at a final concentration of 10%, then incubated at 37 °C for 30 another minutes. The opsonized bacteria were then washed with PBS and incubated with anti-mouse C3 conjugated with FITC (MP Biomedicals) at a final concentration of 1:500 for 30 min. The samples were then washed twice with PBS and resuspended with 200 µL of 2% paraformaldehyde, and fluorescence was determined using a FACS Canto II flow cytometer (BD Biosciences—Franklin Lakes, NJ, USA) and analyzed using the FlowJo software (version 10.10.0).

### 2.8. Biofilm Inhibition Assay

A bacterial stock of BM567 made during the mid-log growth phase was thawed and diluted in LB to an optical density (O.D. _600nm_) of 0.1 and subsequently grown to an O.D. of 0.5. The bacteria were then diluted again to an O.D. of 0.1 and opsonized with normalized, heat-inactivated sera from the immunized and control mice at a final concentration of 0.1 mg/mL. Next, 150 µL of the opsonized bacterial suspension was added to the wells of flat-bottomed 96-well plates, which were incubated statically at 37 °C for 6 h.

After incubation, the supernatant containing the planktonic cells was carefully aspired and discarded, and the biofilm was mechanically removed using a 200 µL pipette tip after the addition of 150 µL of sterile PBS 1× to the wells. The biofilm suspension was dispersed using a vortex and a water bath sonicator with 20 s pulses. The O.D. of the final suspension was measured, and serial dilutions were plated to count CFUs.

### 2.9. Statistical Analysis

The graphs and the statistical analysis were generated using the GraphPad Prism 6 software. The distribution of the values was assessed with the Shapiro–Wilk test, and either parametric or non-parametric tests were chosen based on the distribution of the data. A *p* value < 0.05 was considered statistically significant.

Analysis of IgG production was conducted using two-way ANOVA with Sidak’s post-test. For the antibody binding and complement deposition assays, a statistical analysis was performed using ANOVA, followed by a Dunn’s posttest to assess the differences between control and vaccinated groups in terms of binding capacity and percentage of positive cells, with significance set at *p* < 0.001. The statistical differences in the biofilm inhibition assay and lung colonization were evaluated using the Kruskal–Wallis test.

## 3. Results

### 3.1. Immunization with rFimA and rMrkA Induces a Strong IgG1 Response

Immunization with FimA and MrkA induced specific antibody responses in mouse serum, characterized by a predominance of IgG1 and a low production of IgG2a (Figure 1). MrkA was more immunogenic than FimA, generating three times more antibodies (1898 µg/mL of anti-MrkA versus 640 µg/mL of anti-FimA). The group injected with the protein mix showed a similar pattern, with higher levels of anti-MrkA antibodies and a lower anti-FimA response. The IgG1/IgG2a ratios were also very similar among the different immunization groups, with a predominance of IgG1 and low levels of IgG2a.

The coadministration of MrkA and FimA resulted in similar antibody levels to the group immunized with MrkA alone (1450 µg/mL of anti-MrkA in the co-administered versus 1898 µg/mL in the MrkA group, *p* = 0.17). However, when compared with the FimA-immunized group, the protein mix induced lower levels of FimA antibodies (325 µg/mL of anti-FimA in the co-administered proteins versus 640 µg/mL in the FimA group, *p* = 0.001). This suggests that the combination of Fima and MrkA slightly favors an anti-MrkA response, but a strong anti-FimA response is still mounted.

### 3.2. Antibody Binding onto K. pneumoniae

The ability of FimA and MrkA antisera to bind to *K. pneumoniae* was evaluated using flow cytometry. As shown in Figure 2, both proteins induced antibodies that recognized the native fimbriae at the bacterial surface, with anti-MrkA (Figure 2B,D) showing an increase in binding in comparison with anti-FimA (Figure 2A,D) (88.9%. of FITC-positive cells in the rMrkA group versus 59.6% in the rFimA group, *p* < 0.001). The median fluorescence intensity (MFI) was markedly different between the immunization groups; 254 MFI in the bacteria treated with anti-MrkA sera against 14.2 in those incubated with anti-FimA, *p* < 0.0001). While the coadministration of the proteins did not enhance the binding capacity (88.2% of FITC-positive cells versus 88.9%. In the rMrkA group), the presence of both anti-FimA and anti-MrkA did not interfere with antibody binding in this group (Figure 2C,D).

### 3.3. Vaccine Antibodies Induce Complement Deposition on K. pneumoniae

In accordance with the binding results, antisera from mice immunized with rMrkA and rFimA promoted an increase in C3 deposition on the bacterial surface. As shown in Figure 3, significant antibody-mediated C3 binding was observed in the rMrkA antisera and the co-administration group. While a response was detected in the rFimA antisera, the percentage of positive cells did not reach statistical significance compared to the control group, as determined by analysis of variance (ANOVA) followed by Dunn’s post hoc test (*p* > 0.05). Interestingly, the group immunized with the protein mix elicited antibodies with the strongest effect on C3 deposition (27% of positive cells compared to 18% in the rMrkA group).

### 3.4. Biofilm Formation Is Inhibited in the Presence of Antibodies

The effect of vaccine antibodies on biofilm formation was determined through an in vitro assay by growing the opsonized and control *K. pneumoniae* on polystyrene 96-well flat-bottom plates and harvesting the adhered bacteria. As shown in Figure 4, pre-incubation with sera from mice immunized with MrkA alone or combined with FimA reduced the CFU count of the biofilm recovered from the plate, while anti-FimA sera did not promote a significant reduction in biofilm formation (Figure 4A). Similarly, a reduction in biofilm absorbance is shown in the FimA + MrkA mix group, indicating a negative impact of the antibodies on the biofilm biomass (Figure 4B).

### 3.5. Immunization with rFimA and rMrkA Reduces Lung Colonization by K. pneumoniae

The protective effects of immunization were assessed in a lung colonization model. Immunized and control mice were challenged intranasally with *K. pneumoniae*, euthanized after 48 h and had their lungs removed for CFU counting. The results of this analysis are shown in Figure 5. The co-administration of rFimA and rMrkA reduced lung colonization by *K. pneumoniae* by 80-fold when compared with the control group—an effect that was not observed in the groups injected with the single proteins.

An analysis of the histological slides indicated that, in all challenged groups, the animals exhibited pneumonia, with a predominantly polymorphonuclear diffuse inflammatory infiltrate (Figure 6B–E), in comparison with healthy controls (Figure 6A). Lymphocytes and macrophages were also observed, along with blood vessel congestion and severe hemorrhage, particularly in the challenged control group, where a more pronounced inflammatory infiltrate was present, making it impossible to differentiate the alveolar structures in most of the analyzed fragments (Figure 6B). The inflammation was notably more severe compared to the immunized animals. Dilation and congestion of the vessels surrounding the alveoli were also more abundant in this group.

Among the animals that received the recombinant proteins, the inflammatory infiltrate was more localized in the peribronchiolar region. In the lung tissue of the mice that received rFimA, in addition to polymorphonuclear cells, a significant number of lymphocytes and macrophages was also noted (Figure 6C). The mice immunized with rMrkA presented moderate hemorrhage and vessel congestion in the lungs, but the alveolar structures were identifiable in most analyzed fields (Figure 6D).

In the slide from the animal that received the rFimA and rMrkA mix, the inflammatory infiltrate included polymorphonuclear cells, macrophages, lymphocytes and plasma cells (Figure 6E).

The challenged control also showed higher inflammatory and tissue damage scores in comparison with the immunized and challenged mice (Table 1). More specifically, this group showed increased signs of polymorphonuclear inflammatory infiltrate, loss of alveolar structure, congestion and pulmonary hemorrhage when compared to the mice who were immunized before challenge. As expected, no signs of inflammation or tissue damage were found in healthy, unchallenged controls.

Overall, these findings suggest a reduction in the diffusion and intensity of the inflammatory infiltrate in the lung parenchyma following immunization with rFimA and/or rMrkA compared to the control group.

## 4. Discussion

*K. pneumoniae* is a leading cause of multidrug-resistant infections worldwide [1], reinforcing the need for new therapeutics and/or prophylactic measures. In recent decades, many vaccine candidates have been evaluated; however, there are no vaccines currently available against *K. pneumoniae* pneumonia and systemic diseases.

The present work evaluated the outcomes of immunization with rFimA and rMrkA in a lung colonization model. Following subcutaneous administration, both proteins elicited a strong IgG1 production in serum, although rMrkA was significantly more immunogenic than rFimA, promoting a three-fold increase in antibody levels in comparison with FimA. The combination of rFimA and rMrkA resulted in high antibody levels against each protein included in the mix, although the anti-rFimA response was lower when compared to the group immunized with the isolated protein.

A small IgG2a response was detected, with no differences between the immunization groups. The prevalence of the IgG1 subclass is characteristic of vaccines employing Alum as an adjuvant [22]. In a pulmonary infection model, anti-CPS immunoglobulins of the IgG1 and IgG3 subclasses were shown to be important for lung clearance; however, different defense mechanisms are enhanced based on the IgG isotype. In that work, IgG1 opsonization induced the strongest reduction in bacterial burden and the expression of inflammatory cytokines, while IgG3 was more effective at promoting complement- and neutrophil-mediated bacterial killing [23].

Flow cytometry analysis demonstrated that both anti-rFimA and anti-rMrkA antibodies bind to the *K. pneumoniae* surface, with anti-rMrkA presenting the strongest effect. This result suggests that the recombinant proteins retain the epitopes of the native molecules, eliciting antibodies that can recognize the fimbriae expressed at the bacterium surface. It also demonstrates that the fimbriae—especially the type 3 fimbriae—are accessible for recognition by the immune system, despite the presence of a polysaccharide capsule in this strain.

Adding to the binding results, sera from immunized mice were tested for the ability to activate complement deposition on the bacteria—an important clearance mechanism during infections with Gram-negative pathogens. Sera against rMrkA, but not rFimA, led to a significant increase in C3 deposition on the bacterial surface. The protein mix had the strongest effect, promoting higher levels of C3 attachment to the bacteria. These results confirm the ability of the vaccine antibodies to elicit host responses that contribute to infection control. Since complement activation may cause the direct killing of bacteria through membrane attack complex (MAC) formation, we tested the bactericidal effects of complement through a serum bactericidal assay using normal mouse serum as a complement source (Appendix A). The treatment of opsonized bacteria with complement did not affect survival (Appendix A), suggesting that this strain is resistant to the effects of MAC. Thus, other complement effects, such as opsonization, may contribute to *K. pneumoniae* clearance.

Another important virulence mechanism in *K. pneumoniae* is biofilm formation, which protects the bacteria from antimicrobial agents and host immune defenses, while providing a focus for dissemination within the host [16,24]. Therefore, the ability to impair biofilm formation may contribute to bacterial clearance. In the present study, pre-opsonization with anti-rMrkA serum resulted in reduced biofilm formation in vitro by the *K. pneumoniae* isolate BM567. Similarly, antibodies against the protein mix limited biofilm formation by this strain. Anti-rFimA serum, on the other hand, did not affect biofilm formation. Considering that fimbriae play an essential role in the initial surface adhesion, the colonization of abiotic surfaces could be hindered by blocking the primary structures responsible for adhesion. Attachment to surfaces is the first step in infection establishment, and in ventilator-associated pneumonia, adhesion to endotracheal tubes contributes to colonization and infection of the respiratory tract [17]. Type 3 fimbriae are the main structural components associated with adhesion to abiotic surfaces; however, type 1 fimbriae have also been shown to be important for *E. coli* adhesion to catheters and biofilm formation [25]. The present data suggest a role for anti-MrkA antibodies in inhibiting biofilm formation by *K. pneumoniae*, which could contribute to ameliorate infection in intubated patients within healthcare settings.

Finally, the protective effect of vaccination was evaluated in a mouse model of lung colonization. Mice immunized with the protein mix, but not the individual proteins, showed a significant reduction in bacterial loads in the lungs. This suggests that although each protein alone can elicit strong immune responses, both fimbriae subunits are required for protection against pneumonia. The in vivo data confirm the in vitro functional antibody assays, which show that the co-administration of rFimA and rMrkA induces antibodies that are able to bind to the native fimbriae expressed at the bacterial surface, promote complement deposition and reduce biofilm formation by the bacterium.

While bacterial counts were reduced only in the co-administered group, histopathological analysis indicated that immunization with either rFimA or rMrkA significantly delayed and reduced the intensity of lung congestion and inflammatory infiltration, preserving the alveolar structures. In contrast, the control group exhibited larger areas of damage. Since the exacerbated inflammatory infiltrate directly correlates with greater pulmonary injury, a reduction in inflammation intensity suggests a higher likelihood of resolving infection in the vaccinated mice. Immunization with both rFimA and rMrkA was necessary to decrease the bacterial burden in the lungs. Thus, we hypothesize that both types of fimbriae may have complementary roles in the pathogenesis process.

Studies evaluating the vaccine potential of fimbriae have found variable results. Extracted fimbriae were found to be immunogenic and protective in a mouse pneumonia model [26]. Anti-MrkA mAbs were shown to reduce biofilm formation, increase opsonophagocytosis, reduce lung colonization and extend survival in immunized mice [27,28,29]. Similarly, recombinant MrkA (rMrkA) in either its monomeric or oligomeric forms also reduced lung colonization [27]. The fimbrial adhesin MrkD and derived peptides also obtained encouraging results in a lethal pneumonia/sepsis model [30,31].

While type 3 fimbriae and its subunits received the most attention as vaccine candidates, type 1 fimbriae also play an essential role during colonization and infection. The major subunit of the type 1 fimbriae (FimA) was tested in a lethal infection model with the HMV+ strain. Although FimA alone was not able to protect the mice; it reduced the bacterial burden in the lungs and increased opsonophagocytic killing. Conversely, a chimeric protein containing FimA was protective against fatal infection with different clinical strains of *K. pneumoniae* [32].

FimA was also evaluated as a candidate in *Porphyoromas gingivalis* [33,34,35] and *Acinetobacter baumannii* vaccines [36] and used as an adjuvant in formulations against *Edwardsiella tarda* [37]. Similarly, recombinant FimH from *Vibrio parahaemolyticus* was immunogenic and protective in a lethal infection model [38]. Meanwhile, FimH from *Salmonella typhimurium* [39] and *E. coli* [40] were successfully tested as adjuvants due to their ability to interact with mucosal surfaces.

Overall, the use of either complete fimbriae or its subunits has shown promising results in many different bacterial species, including *K. pneumoniae*. Due to the adhesive properties of the fimbriae, blocking its ability to attach to surfaces could limit colonization of the host mucosa or abiotic surfaces, such as catheters and endotracheal tubes.

Despite the encouraging results, the present study has a few limitations. It lacks data on the longevity of the vaccination response, which is important when evaluating the length of protection conferred by the vaccine. Also, the intrinsic differences in the murine immune response when compared to humans have been correlated with failure in the progression from pre-clinical trials to the clinical evaluation of new therapies [41]. Such differences may be evident during the infection evolution. While there is an overlap between mice and human immune responses, mice are naturally highly resistant to *K. pneumoniae* infections, in a strain-dependent manner [42]. More information could be acquired in future clinical trials; however, so far, there are no reports of trials evaluating fimbrial subunits of *K. pneumoniae*.

## 5. Conclusions

In conclusion, the present work has demonstrated that immunization with rFimA and rMrkA can aid in the control of *K. pneumoniae* infections by reducing lung colonization and inhibiting adhesion to abiotic surfaces, a crucial step in the development of healthcare-associated infections. It also shows that the two fimbrial components are required for protection against pneumonia, suggesting that these molecules have complementing roles in bacterial pathogenesis.

## Figures and Tables

**Figure 1 vaccines-13-00303-f001:**
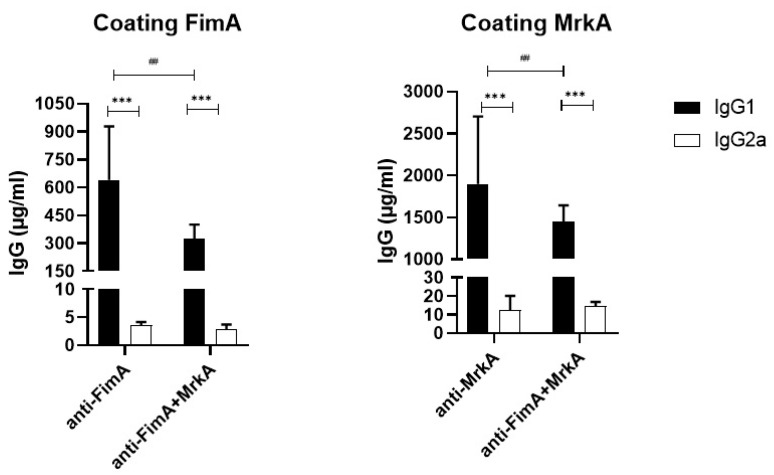
Serum antibody production in mice immunized with MrkA and/or FimA. The concentration of IgG1 and IgG2a is shown for each immunization group (FimA, MrkA and FimA + MrkA). Comparisons between the groups were performed using two-way ANOVA with Sidak’s post-test. *** *p* < 0.0001 when comparing IgG1 × IgG2a responses. ## *p* < 0.01 when comparing antibody levels among different immunization groups.

**Figure 2 vaccines-13-00303-f002:**
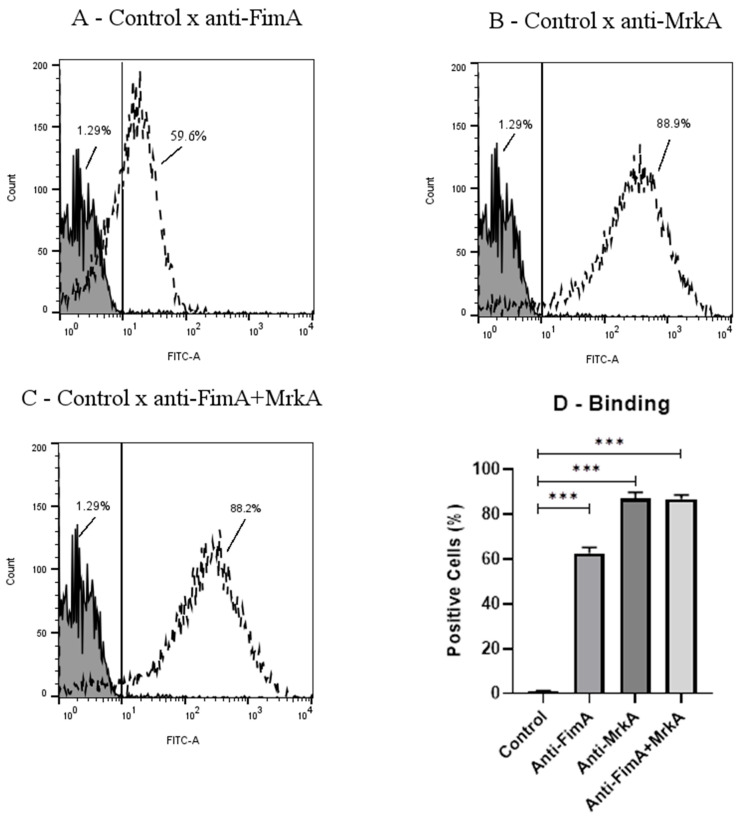
Binding of anti-FimA and anti-MrkA antisera to K. pneumoniae. The percentages of FITC-positive cells are shown for each group. The solid lines represent bacteria incubated with control sera, while the dashed lines indicate bacteria treated with sera from vaccinated mice. (**A**) control x anti-FimA; (**B**) control x anti-MrkA; (**C**) control x anti-FimA + MrkA. (**D**) Fluorescence comparison among immunization groups. Statistical differences between control and vaccinated groups were evaluated using ANOVA with a Dunn’s post-test (*** p < 0.001) in comparison with control.

**Figure 3 vaccines-13-00303-f003:**
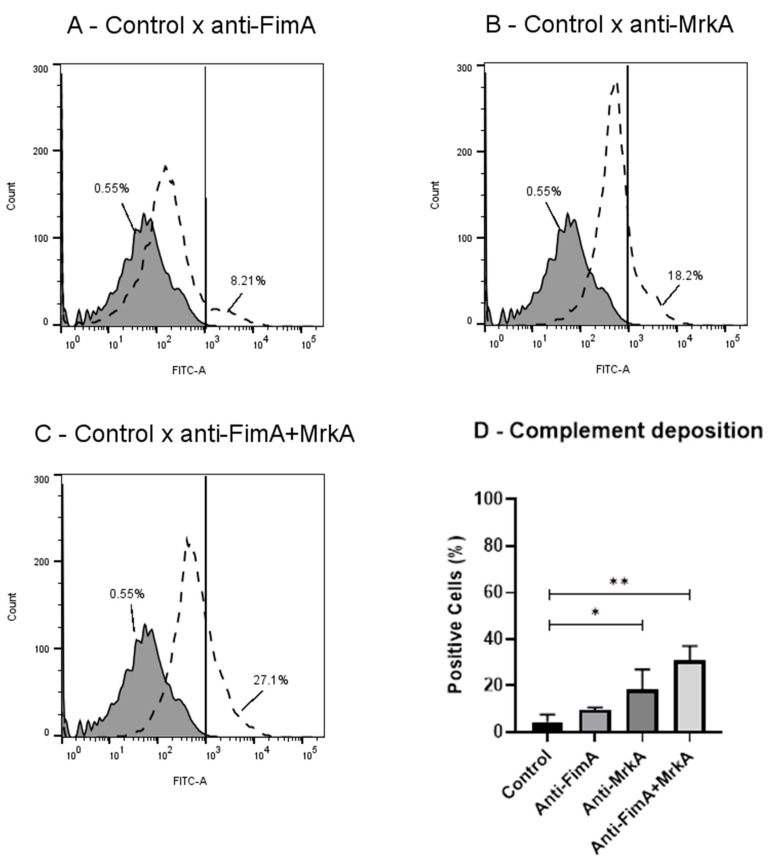
Antibody-mediated complement deposition. *K. pneumoniae* was incubated in the presence of sera from control or vaccinated mice, followed by the addition of NMS as a complement source. Detection was performed using FITC-conjugated anti-mouse C3. The percentages of FITC-positive cells are shown for each group. The solid lines represent C3 deposition on bacteria incubated with control sera, while the dashed lines indicate those treated with sera from vaccinated mice. (**A**) control × anti-FimA; (**B**) control × anti-MrkA; (**C**) control × anti-FimA + MrkA. Fluorescence comparison among immunization groups. (**D**) Fluorescence comparison among immunization groups. Statistical differences between control and vaccinated groups were evaluated using ANOVA with a Dunn’s post-test (* *p* < 0.05 and ** *p* < 0.01) in comparison with control.

**Figure 4 vaccines-13-00303-f004:**
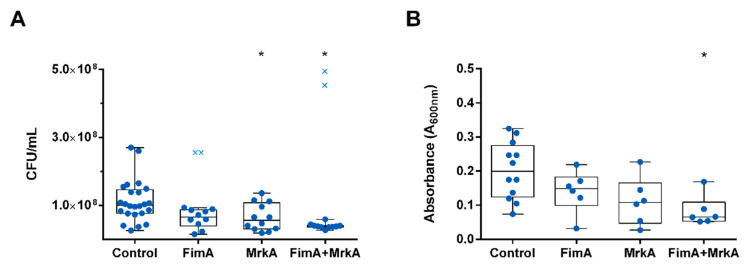
Biofilm formation is inhibited by vaccine antibodies. (**A**) CFU counts of biofilm formed in the presence of sera from control and vaccinated mice. (**B**) Absorbance (A_600nm_) of the biofilms recovered from the plate. Statistical differences were evaluated via Kruskal–Wallis test (* *p* < 0.05). The × symbol indicates the statistical outliers removed from the analysis.

**Figure 5 vaccines-13-00303-f005:**
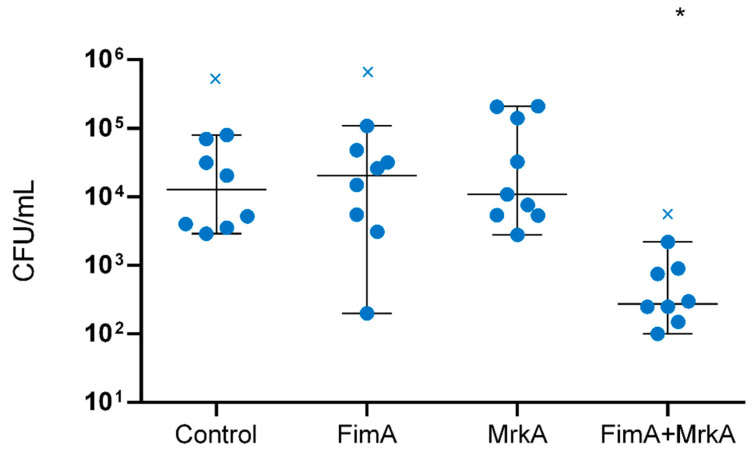
Lung colonization by *K. pneumoniae* in vaccinated × control mice. CFU counts in the lung homogenates of control and vaccinated mice were determined 48 h after intranasal challenge with *K. pneumoniae* BM567. Each dot represents one animal. Statistical differences were evaluated via Kruskal–Wallis test. The × symbol indicates the statistical outliers removed from the analysis. * *p* < 0.05 in comparison with the control.

**Figure 6 vaccines-13-00303-f006:**
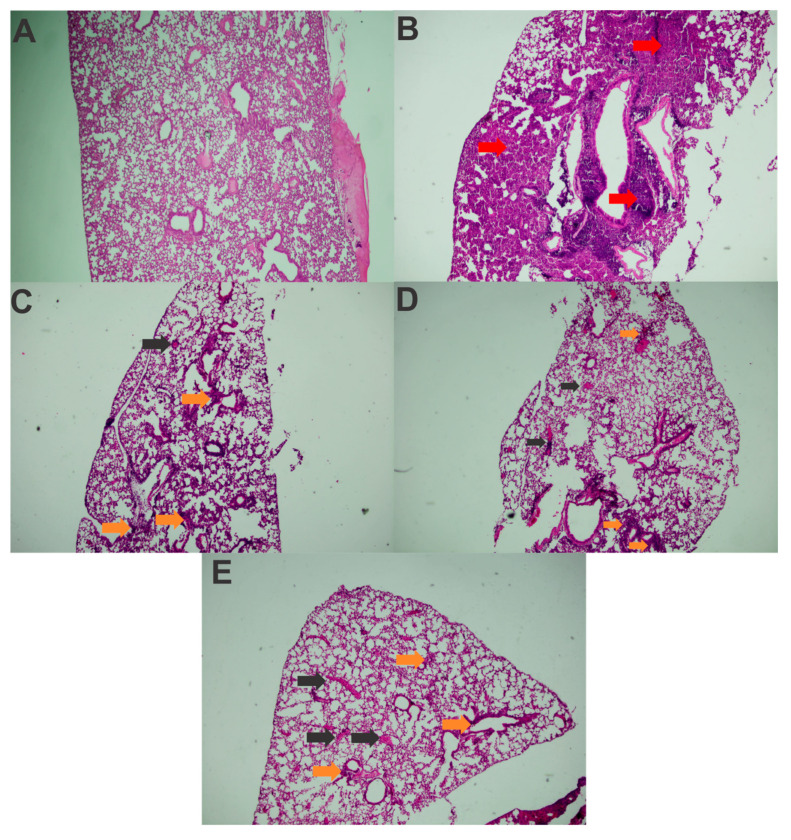
Histopathological analysis of the lung tissue from mice before and after challenge with *K. pneumoniae* (40× amplification). (**A**) Healthy unchallenged control; (**B**) challenged control; (**C**) Immunized with rFimA and challenged; (**D**) immunized with rMrkA and challenged; (**E**) immunized with rFimA + rMrkA and challenged. The red and orange arrows indicate intense and moderate inflammatory infiltrates, respectively; black arrows, vessel congestion.

**Table 1 vaccines-13-00303-t001:** Histopathological evaluation of the lungs of the animals infected with the BM567 strain.

	Healthy Control	Challenged Control	FimA	MrkA	FimA + MrkA
Polymorphonuclear Inflammatory Infiltrate	-	+++	++	++	++
Mononuclear Inflammatory Infiltrate	-	++	++	+	++
Alveolar Structure Loss	-	++	+	+	++
Congestion	-	+++	++	++	++
Edema	-	+	+	+	+
Pulmonary hemorrhage	-	+++	++	++	++

The histopathological findings were classified as absent (-), slight (+), moderate (++) or intense (+++).

## Data Availability

The data presented in this study are available on request from the corresponding author.

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
