# Peer review of "Protection Against Pneumonia Induced by Vaccination with Fimbriae Subunits from Klebsiella pneumoniae"

_vaccines, 2025, doi:10.3390/vaccines13030303_

Round 1
Reviewer 1 Report
Comments and Suggestions for Authors
Review of “Protection against …. from Klebsiella pneumoniae.”
In this article the authors have used two fimbriae proteins FimA and MrkA from Klebsiella pneumoniae as targets to raise vaccines against this pathogen and tested its efficacy for binding to Klebsiella in vitro, to inhibit biofilm formation and to inhibit infection in vivo in lung tissue of mice. They demonstrate that both proteins can be used as potential targets for vaccine development, more so MrkA. Although the evidence is compelling in experiments in vitro and in vivo, a few significant clarifications need to be addressed, particularly, important controls in the in vivo experiments, to fully stake the claim that FimA and MrkA can be used as viable targets for vaccine development. A revised version of the manuscript is worthy of consideration for publication in Vaccines.
Specific comments:
1. It is not clear from the data presented in Figures 2 and 3 if the authors used equivalent amounts of antibodies or total serum proteins (0.06mg/mL protein as stated in the methods section), is this the total serum protein used or antibody concentration? If the amounts indicated are total serum proteins, then the result is per expectation since FimA antibodies were produced at a lower level (Figure 1) and hence the lower level of FITC positive cells. Was the experiment done with identical level of FimA and MrkA antibodies and total serum protein adjusted using non-immunized serum? If this is the case, then it must be stated clearly in the methods section with the exact amount of antibody used.
2. It is understandable the authors generated both antibodies in mice, but to get a better understanding of the efficacy of the two antibodies in question in vitro, it would have been ideal to generate one of the antibodies in a different host (rabbit) to effectively monitor the two antibodies separately in the mixed antibodies experiments (Fig 2C & 3C) by using secondary antibodies that are labelled with spectrally distinct fluorophores. Such an approach would have yielded more valuable information. For example, (1) the authors could have inferred the precise levels of binding of each antibody in the mixed experiment and (2) changed the order of addition of the antibodies to understand if binding of one interfered with binding of the other. This would have helped the authors interpret the in vivo mixed experiments data better (Figures 5).
3. Although the in vivo results presented in Figure 5 are impressive and the responses observed significant, two very significant controls are missing making it less quantifiable. Histopathological analysis of unchallenged lung tissue from control and immunized mice must be presented to more accurately assess the efficacy of immunization. How do the authors know the difference between the data in panel D vs unchallenged immunized lung? Is the efficacy of immunization at 10% or 50% or some other number? It is not possible to evaluate this without an unchallenged immunized control, which is lacking in the experiment, ultimately a result that will be pivotal for making this a likely candidate for vaccine development. The authors should attempt to quantify the immunogenic efficacy to combat infection, perhaps score for infection specific markers in specific cell types of the lung.
4. Results section 3.1 - 3.3: Please refer to specific figure panels when describing the results in the text. Figures 2 and 3, label the individual figure panels appropriately with sample names for ease of reading and comparing the results.
Author Response
Reviewers’ response
We would like to thank the reviewers for the careful analysis of the manuscript. Based on the comments, the manuscript has been revised, with all the modifications highlighted in yellow. New experiments were also performed. We have replied to each comment individually as follows:
Reviewer 1
In this article the authors have used two fimbriae proteins FimA and MrkA from Klebsiella pneumoniae as targets to raise vaccines against this pathogen and tested its efficacy for binding to Klebsiella in vitro, to inhibit biofilm formation and to inhibit infection in vivo in lung tissue of mice. They demonstrate that both proteins can be used as potential targets for vaccine development, more so MrkA. Although the evidence is compelling in experiments in vitro and in vivo, a few significant clarifications need to be addressed, particularly, important controls in the in vivo experiments, to fully stake the claim that FimA and MrkA can be used as viable targets for vaccine development. A revised version of the manuscript is worthy of consideration for publication in Vaccines.
Specific comments:
1. It is not clear from the data presented in Figures 2 and 3 if the authors used equivalent amounts of antibodies or total serum proteins (0.06mg/mL protein as stated in the methods section), is this the total serum protein used or antibody concentration? If the amounts indicated are total serum proteins, then the result is per expectation since FimA antibodies were produced at a lower level (Figure 1) and hence the lower level of FITC positive cells. Was the experiment done with identical level of FimA and MrkA antibodies and total serum protein adjusted using non-immunized serum? If this is the case, then it must be stated clearly in the methods section with the exact amount of antibody used.
The experiments were performed with normalized sera. To perform the normalization, the more concentrated serum (anti-MrkA) was diluted in control serum to contain the same antibody concentration as anti-FimA. 0.06mg/mL refers to the final concentration of antibodies in the sample, not of total proteins. This sentence was adjusted in the manuscript to clarify that information (lines 177-178)
2. It is understandable the authors generated both antibodies in mice, but to get a better understanding of the efficacy of the two antibodies in question in vitro, it would have been ideal to generate one of the antibodies in a different host (rabbit) to effectively monitor the two antibodies separately in the mixed antibodies experiments (Fig 2C & 3C) by using secondary antibodies that are labelled with spectrally distinct fluorophores. Such an approach would have yielded more valuable information. For example, (1) the authors could have inferred the precise levels of binding of each antibody in the mixed experiment and (2) changed the order of addition of the antibodies to understand if binding of one interfered with binding of the other. This would have helped the authors interpret the in vivo mixed experiments data better (Figures 5).
We thank the reviewer for the interesting suggestion. However, our animal facility does not have the proper structure to house SPF rabbits, so we could not perform the experiments safely. Furthermore, it is possible that the antibody response may vary in different hosts, therefore interfering with the comparison. Although knowing the precise levels of antibody binding to each protein in the mix would provide valuable information on their contribution to the immune response, the fact that the protein mix still induced high antibody levels against each protein by ELISA suggests that the combined vaccine is an interesting approach, able to elicit protective immune responses.
3. Although the in vivo results presented in Figure 5 are impressive and the responses observed significant, two very significant controls are missing making it less quantifiable. Histopathological analysis of unchallenged lung tissue from control and immunized mice must be presented to more accurately assess the efficacy of immunization. How do the authors know the difference between the data in panel D vs unchallenged immunized lung? Is the efficacy of immunization at 10% or 50% or some other number? It is not possible to evaluate this without an unchallenged immunized control, which is lacking in the experiment, ultimately a result that will be pivotal for making this a likely candidate for vaccine development. The authors should attempt to quantify the immunogenic efficacy to combat infection, perhaps score for infection specific markers in specific cell types of the lung.
We appreciate the suggestion presented by the reviewer. Since it is highly unlikely that subcutaneous immunization with the recombinant proteins would induce significant changes in the mouse lungs, we have added unimmunized and unchallenged mouse lungs as a healthy control in the study (Figure 6A). These samples represent healthy lungs and were compared to unimmunized + challenged mice and immunized + challenged mice. The results were presented as a histopathological score based on inflammation and tissue damage signs found in the slides and shown in Table 1. The analysis showed that there was a considerable reduction in the presence of inflammatory infiltrate, alveolar damage and congestion in the immunized animals when compared to the sham-immunized after challenge, while unchallenged healthy controls displayed no signs of inflammation nor tissue damage. While the signs were not completely absent in the vaccinated animals, the results indicate that the reduced lung tissue damage correlated with lower bacterial loads in the lungs and better prognosis. This information was added to the methods and results sections (lines 156-160 and 345-350)
4. Results section 3.1 - 3.3: Please refer to specific figure panels when describing the results in the text. Figures 2 and 3, label the individual figure panels appropriately with sample names for ease of reading and comparing the results.
The figure panels were labeled and cited individually when described in the text (lines 252-260)
Reviewer 2 Report
Comments and Suggestions for Authors
My concern about this manuscript is given below:
The manuscript should provide more detailed information about the vaccination protocol, including the number of doses, timing between immunizations, and adjuvant details.
The study mentions complement activation and biofilm inhibition but does not explore the underlying mechanisms. For example, additional data on opsonophagocytosis or specific molecular interactions would strengthen the findings.
The manuscript relies heavily on descriptive statistics. Including statistical significance tests for all comparative results, especially for biofilm inhibition and complement activation, would improve the rigor.
The authors should acknowledge the limitations of their study, such as the lack of human clinical data and potential differences in immune responses between mice and humans.
Figures such as histological slides and flow cytometry plots should include high-resolution images and more comprehensive labeling for better clarity.
The manuscript lacks data on the longevity of the immune response post-vaccination. Adding a time-course study would strengthen the conclusions.
In general, IgG1 is typically associated with Th2-biased responses; how might this affect the vaccine’s efficacy in terms of complement activation and bacterial clearance?
Why does co-administration of rFimA and rMrkA reduce the anti-FimA response? Is there any evidence of antigen interference or competition when both proteins are administered together?
What is the significance of biofilm inhibition in clinical contexts? Given that biofilm formation is a key virulence factor in healthcare-associated infections, how does the vaccine perform against biofilms on actual medical devices?
Comments on the Quality of English Languageno
Author Response
We would like to thank the reviewers for the careful analysis of the manuscript. Based on the comments, the manuscript has been revised, with all the modifications highlighted in yellow. New experiments were also performed. We have replied to each comment individually as follows:
Reviewer 2
My concern about this manuscript is given below:
1. The manuscript should provide more detailed information about the vaccination protocol, including the number of doses, timing between immunizations, and adjuvant details.
Detailed information regarding number of doses, timing between immunizations, and adjuvant was added and can be found in lines 140 to 143.
2. The study mentions complement activation and biofilm inhibition but does not explore the underlying mechanisms. For example, additional data on opsonophagocytosis or specific molecular interactions would strengthen the findings.
As suggested, we have performed two OPA assays using isolated human neutrophils and different concentrations of sera (2, 5 and 10%). However, we could not see any effect of the vaccine antibodies on phagocytosis. Since this experiment requires further standardization of serum concentrations, adjustment of bacteria/cells ratio and optimization of the neutrophil isolation protocols, we do not feel confident to present these results as our final analysis. We will proceed to establish a protocol for evaluating the effect of vaccine antibodies o bacterial phagocytosis, to be published in the future.
Figure 1: In vitro phagocytosis of K. pneumoniae in presence of sera from immunized mice. Bacteria were opsonized with sera from mice vaccinated with rFimA, rMrkA or rFimA+rMrkA, followed by incubation with isolated human neutrophils. The control group was incubated with sera from sham immunized mice. The number of bacteria surviving treatment is shown for each group. Comparison between groups was conducted by ANOVA with Dunn’s posttest. ns=not significant
Figure 2. In vitro phagocytosis of K. pneumoniae in presence of anti-rMrkA antibodies. Bacteria were opsonized with sera from mice vaccinated with rMrkA at 5 and 10%, followed by incubation with isolated human neutrophils. The control group was incubated with sera from sham immunized mice. The number of bacteria surviving treatment is shown for each group. Comparison between groups was conducted by ANOVA with Dunn’s posttest. ns=not significant
3. The manuscript relies heavily on descriptive statistics. Including statistical significance tests for all comparative results, especially for biofilm inhibition and complement activation, would improve the rigor.
As suggested, statistical tests have been included for the experiments. We have added Figure 2D to the Results section, comparing the binding capacity with the presence of both anti-FimA and anti-MrkA. The figure legend has been updated to include: "D) Fluorescence comparison among immunization groups. Statistical differences between control and vaccinated groups were evaluated using ANOVA with a Dunn’s posttest (***p<0.001) in comparison with control."
Additionally, we have included Figure 3D in the Results section, which compares the percentage of positive cells. Statistical significance was evaluated in the same manner, and the appropriate tests were performed to enhance the rigor of the analysis.graphs were added with the adequate statistical analysis.
4. The authors should acknowledge the limitations of their study, such as the lack of human clinical data and potential differences in immune responses between mice and humans.
A paragraph discussing the study limitations has been added to the manuscript (lines 503-512).
5. Figures such as histological slides and flow cytometry plots should include high-resolution images and more comprehensive labeling for better clarity.
Figures 2, 3 and 6 have been replaced with higher resolution images.
6. The manuscript lacks data on the longevity of the immune response post-vaccination. Adding a time-course study would strengthen the conclusions.
The manuscript does not include a longitudinal study to evaluate the durability of the immune response post-vaccination. However, recombinant protein vaccines formulated with aluminum-based adjuvants have been shown to enhance antibody-mediated immunity for extended periods (Laera, HogenEsch, O'Hagan, 2023 https://doi.org/10.3390/pharmaceutics15071884). Furthermore, a study by Gupta et al. (2020 – doi: 10.3389/fimmu.2020.00988) demonstrated that mice immunized with a recombinant fusion protein (LcrV-HSP70) adjuvanted with alum sustained a protective immune response for at least 60 days post-immunization, as evidenced by protection against bacterial challenge during this period.
In summary, we acknowledge the importance of a longitudinal study to evaluate the duration of the induced response but understand that the current data is sufficient to support future studies using this formulation, which will include analysis of the immune responses over time.
7. In general, IgG1 is typically associated with Th2-biased responses; how might this affect the vaccine’s efficacy in terms of complement activation and bacterial clearance?
This is a very interesting point raised by the reviewer. As discussed by Lilienthal et al. (2018), IgG1 has limited capacity to activate the classical complement pathway and a reduced affinity for Fc activator receptors, resulting in decreased opsonophagocytosis. This reduced functionality can impair the vaccine's ability to effectively eliminate pathogens. However, previous data from our group and others has shown that vaccines composed of recombinant proteins using Alunm as adjuvant, even though inducing strong IgG1 responses, are still able to promote complement deposition on the bacterial surface and favor clearance by phagocytosis, which is in accordance with the current data. (Converso TR, 2017 - doi: 10.1016/j.vaccine.2017.08.010; Goulart et al, 2017 - doi: 10.1016/j.micpath.2017.06.004;
Furthermore, in a study investigating anti-CPS IgG subclasses of K. pneumoniae, IgG3 showed better affinity and higher complement deposition and killing rates. However, it was also observed that IgG1 diminished lung colonization, in addition to an increase in phagocytosis when compared to IgG3 (Motley et al., 2020 - doi: 10.1128/mBio.02059-20).
Overall, a balance between the IgG isotypes is needed to ensure an efficient humoral immune response following immunization. Therefore, in future studies, we intend to investigate other adjuvants and vaccine strategies that aim to improve the IgG2a/IgG1 ratio.
8. Why does co-administration of rFimA and rMrkA reduce the anti-FimA response? Is there any evidence of antigen interference or competition when both proteins are administered together?
There is no evidence in the literature that coadministration of FimA and MrkA could have an inhibitory effect on antibody production. However, to our knowledge, this is the first study evaluating the coadministration of the two recombinant fimbriae subunits. It is important to notice that, while some reduction in the IgG production and in the FITC intensity values was observed, no differences were seen in the percentage of FITC+-cells in the flow cytometry assays using the sera from the co-administered group. Although we can not point out the reason for the reduction in antibody levels in the FimA+MrkA group, this result did not affect the ability of the mixed vaccine to protect against pneumonia, even suggesting a higher quality of the antibody response in this group, leading to a reduction in bacterial loads in the lungs.
9. What is the significance of biofilm inhibition in clinical contexts? Given that biofilm formation is a key virulence factor in healthcare-associated infections, how does the
From a clinical perspective, the ability of vaccine antibodies to inhibit biofilm formation could predict a protective effect of these antibodies in limiting bacterial adherence to medical devices as catheters, which are common sources of hospital acquired infections in intensive care patients. While we did not evaluate biofilm formation in actual medical devices, previously, a E. coli in-vitro colonization model made inside a chambered device, using artificial and human urine and a Foley catheter showed that the expression of type 1 fimbriae is essential to colonization of the silicone device (Reisner et al., 2014 - doi: 10.1128/JB.00985-13).Previous studies have also indicated that fimbriae contribute to adhesion and biofilm formation on host tissues. Although we have not tested this hypothesis in the present study, it is possible that the limiting effects of anti-fimbriae antibodies on biofilm formation could diminish the bacterial ability to colonize host tissues and promote infection.
Round 2
Reviewer 1 Report
Comments and Suggestions for Authors
The authors have appropriately revised the manuscript to address the suggested comments
Reviewer 2 Report
Comments and Suggestions for Authors
The current manuscript is improved enough; no comments required now